# Nicotine Replacement Therapy during Pregnancy and Child Health Outcomes: A Systematic Review

**DOI:** 10.3390/ijerph18084004

**Published:** 2021-04-11

**Authors:** Julie Blanc, Barthélémy Tosello, Mikael O. Ekblad, Ivan Berlin, Antoine Netter

**Affiliations:** 1Department of Obstetrics and Gynecology, North Hospital, APHM, Chemin des Bourrely, 13015 Marseille, France; antoine.netter@ap-hm.fr; 2EA3279, CEReSS, Health Service Research and Quality of Life Center, Aix-Marseille University, 13284 Marseille, France; 3Department of Neonatology, North Hospital, Assistance Publique des Hôpitaux de Marseille, 13015 Marseille, France; Barthelemy.TOSELLO@ap-hm.fr; 4CNRS, EFS, ADES, Aix-Marseille University, 13284 Marseille, France; 5Department of General Practice, Institute of Medicine, Turku University, Turku University Hospital, 20014 Turku, Finland; moekbl@utu.fi; 6Department of Pharmacology, Assistance Publique-Hôpitaux de Paris, Sorbonne University, 75013 Paris, France; ivan.berlin@aphp.fr; 7University Centre of General Medicine and Public Health, 1011 Lausanne, Switzerland; 8Institut Méditerranéen de Biodiversité et d’Écologie Marine et Continentale (IMBE), Aix Marseille University, CNRS, IRD, Avignon University, 13003 Marseille, France

**Keywords:** smoking cessation agents, tobacco use cessation devices, pregnancy, congenital abnormalities, child health, respiration disorders, attention deficit disorder with hyperactivity

## Abstract

Tobacco smoking in pregnancy is a worldwide public health problem. A majority of pregnant smokers need assistance to stop smoking. Most scientific societies recommend nicotine replacement therapy (NRT) during pregnancy but this recommendation remains controversial because of the known fetal toxicity of nicotine. The objective of this systematic review was to provide an overview of human studies about child health outcomes associated with NRT use during pregnancy. The electronic databases MEDLINE, the Cochrane Database, Web of Science, and ClinicalTrials.gov were searched from the inception of each database until 26 December 2020. A total of 103 articles were identified through database searching using combination of keywords. Out of 75 screened articles and after removal of duplicates, ten full-text articles were assessed for eligibility and five were included in the qualitative synthesis. NRT prescription seems to be associated with higher risk of infantile colic at 6 months as in case of smoking during pregnancy, and with risk of attention-deficit/hyperactivity disorder. No association between NRT during pregnancy and other infant health disorders or major congenital anomalies has been reported. Well-designed controlled clinical trials with sufficient follows-up are needed to provide more information on the use of NRT or other pharmacotherapies for smoking cessation during pregnancy on post-natal child health outcomes.

## 1. Introduction

Maternal smoking during pregnancy is an important and preventable major risk factor of adverse obstetrical and neonatal outcomes including placental abruption, miscarriage, stillbirth, preterm birth, and low birthweight [1]. Tobacco smoking in pregnancy is a worldwide public health problem and is estimated to be 1.7% (95% CI, 0.0–4.5) [2]. Prevalence of maternal smoking ranges between 10.9% and 38.4% in Europe with major between country disparities [2].

Cessation of smoking during pregnancy benefits maternal and child health [3]. Many pregnant smokers succeed in stopping smoking spontaneously; this is the situation of 54% of Finnish pregnant smokers, 41.6% of French pregnant smokers, 41.9% of pregnant smokers in four American states, and less than 35% of Australian women [4,5,6,7]. Hence, a majority of pregnant smokers need assistance to stop smoking. Nicotine replacement therapy (NRT) has been demonstrated to help non-pregnant smokers to stop [8]. In pregnant smokers, literature data show that the prescription of NRT during pregnancy is not associated with smoking cessation during pregnancy or at the end of pregnancy if one considers randomized placebo controlled studies versus placebo [9,10]. However, based on the analysis of all available studies, the prescription of NRT during pregnancy is associated with smoking cessation during pregnancy and at the end of pregnancy [9].

Nicotine is a neurotoxin and may affect development of fetal nerve tissues and brain development [11,12]. The use of NRT during pregnancy has been controversial because of potential long-term pulmonary consequences based on animal studies [13]. Cognitive impairments after prenatal exposure to nicotine have been demonstrated in animals and seem to stretch into adulthood [14,15]. NRT have been classified by the United States Food and Drug Administration as pregnancy category C or D depending on the galenic form (i.e., potential benefits may warrant use of the drug in pregnant women despite potential risks) [16]. NRT can be delivered as a buccal absorption product: gum, lozenge, inhaler, nasal and buccal spray, or as a patch with transdermal delivery of nicotine. As of today, there is no evidence of increased risk of pregnancy or birth outcomes associated with the use of NRT [17,18].

Therefore, the general view is that NRT during pregnancy is safer than smoking [19]. Several scientific societies in obstetrics and gynecology have provided guidelines in favor of the prescription of NRT in pregnant smokers who failed to quit without NRT [20,21,22].

Despite several systematic reviews and recommendations on the benefit-risk ratio of NRT during pregnancy, current knowledge is insufficient about health outcomes in children whose mother used NRT for smoking cessation in pregnancy. Therefore, the objective of this study was to provide a systematic review of published studies about child health outcomes associated with NRT administered during pregnancy.

## 2. Materials and Methods

### 2.1. Search Strategy

This systematic review was performed according to Preferred Reporting Items for Systematic Reviews and Meta-Analyses (PRISMA) guidelines [23]. The electronic databases MEDLINE, the Cochrane Database, Web of Science, and ClinicalTrials.gov were searched from the inception of each database until 26 December 2020. The following keywords were used: “Tobacco Use Cessation Devices” [Mesh] AND “Pregnancy” [Mesh].

### 2.2. Eligibility Criteria and Main Outcome Measures

No restrictions on language, and publication status were applied. Abstracts were included if there was sufficient information to assess the study’s quality. Original studies that explored NRT during pregnancy and dealing with outcomes in childhood were eligible for inclusion. The exclusion criteria were: non-human research, review/meta-analysis/summary, opinion/letter/response, case reports, recommendations, articles with no data on the child health, and articles reporting only data on birth and neonates (before 28 days). The primary outcome of this analysis were data on postnatal health outcomes after 28 days.

### 2.3. Data Extraction

Two reviewers (JB and AN) conducted data extraction independently and then discussed to reach a consensus. The very first selection of the papers was made on the basis of papers title and abstract, and eligible ones were selected for full text review. All variables of interest concerning the studies and publications were extracted using a form. The extracted data were authors’ names, the year of publication, the type of study, the sample size, the period of inclusion, the country of the study, and the article main outcomes.

## 3. Results

A total of 103 articles were identified through database searching using combination of keywords. Out of 75 screened articles and after duplicates removed, ten full-text articles were assessed for eligibility and five included in qualitative synthesis (Figure 1).

The studies included in the present review were conducted in Denmark (two studies) and in the United Kingdom (three studies). These papers reported different outcomes. One study investigated infantile colic [24], two studies infant development impairment and reported respiratory problems [25,26], one major congenital anomalies [27], and one attention-deficit/hyperactivity disorder [28].

The five articles included in qualitative synthesis are summarized in Table 1.

### 3.1. Infantile Colic

In 2012, Milidou et al. published a study with the aim to examine whether the association between smoking and infantile colic is due to nicotine [24]. This was based on data from the Danish National Birth Cohort, a nationwide population-based cohort of pregnant women between 1996 and 2002. The Danish pregnant women were invited by their general practitioner to join the cohort at the first antenatal visit. The data were collected from maternal interviews. Exposures during pregnancy were assessed through computer-assisted telephone interviews at second- and third-trimester (around 17 and 32 weeks of gestation). After combining mothers’ replies, the women were allocated to one of 4 mutually exclusive categories: NRT users, smokers, smokers using NRT (combination), and unexposed during pregnancy. Maternal smoking but not NRT use during the postnatal period was assessed in the postpartum interview. Using the interview of women, they evaluated the infant’s behavior, development, nutrition, and frequency and duration of cry episodes and then when the child was 6 months old. Infantile colic cases were identified based on the modified Wessel’s criteria: crying or fussing for more than 3 h a day for more than 3 days a week. The first crying period had to start before the age of 3 months and had to be unrelated to teeth cutting or any recognized disease. Among the 63,128 infants (46,660 infants unexposed to nicotine, 207 exposed to NRT, 15,016 to smoking, and 1245 to a combination of smoking and NRT), 4974 (7.9%) infants fulfilled the criteria for infant colic. The adjusted Odds Ratio (aOR) for infantile colic among infants exposed prenatally, and not postnatally, to NRT was similar to that of infants exposed prenatally to tobacco smoke (aOR, 1.6, 95 CI 1.0–2.5, reference: infants unexposed to nicotine and/or smoking). These results were adjusted for maternal age, first parity, daily coffee consumption, weekly consumption, and binge-drinking episodes. This study provided similar results with several models adjusted or unadjusted for socioeconomic status and other potential confounders. In this cohort, among the NRT users, most used one type of NRT (*n* = 194, 93.7%), and a minority used two different types (*n* = 13, 6.3%), inhalator and gum being the most popular combination. In most cases, NRT users preferred short-release products, and almost half of NRT users continued active smoking (combined smoking and slow-release NRT in most cases).

### 3.2. Infant Impairment (Disability and/or Behavior and Development Problems)

The Smoking, Nicotine And Pregnancy (SNAP) trial was one of the few randomized controlled trial (RCT) assessing the efficacy of NRT during pregnancy; it included 521 women assigned to receive nicotine patches (15 mg per 16 h) or placebo patches (*n* = 529) [29]. This study also provided data on 2-years-old infants born to participating women (follow-up of 888 infants of the 1010 singleton neonates) in seven antenatal hospitals in the Midlands and North-West England [25,30]. The prespecified outcomes were disability behavior and development, and respiratory symptoms in infants at 2 years of age. These outcomes were evaluated by a postal questionnaire. This 24-month participant questionnaire included five domains of the Ages and Stages Questionnaire 3rd edition (ASQ-3) (communication, gross motor, fine motor, problem solving, and personal-social development); seven additional items investigated general and specific parental concerns about infant development; and Health professional questionnaire. Infant impairment was defined as presence of disability and/or behavioral and development problem(s) and categorized as “survival with no impairment”, “survival with definite impairment” and “survival with suspected impairment”. This categorization was based on responses to the previous questionnaires sent to participants. Infants were classed as having a respiratory problem thanks to dedicated item on the 24-month participant questionnaire or Health professional questionnaire. As a reminder: this RCT did not show a significant difference, at delivery and after delivery, in the biochemically validated prolonged smoking cessation rates between the nicotine and the placebo patch groups. In a secondary analysis of the SNAP data with 88% of follow-up to 2 years, Cooper et al. found that 72.6% of infants whose mother was assigned to the nicotine patch group had survived with no impairment, compared with 65.5% of infants born to participants in the placebo group (OR 1.40, 95% CI 1.05 to 1.86). Similar results were found from different analyses (complete case analysis adjusted for clustering by twin pregnancies, and multiple imputation intention-to-treat analysis). The authors reported no significant difference for respiratory disorders (including hospital admission for respiratory problems and/or problems with chest or breathing and/or wheeze or whistling in chest and/or doctor diagnosed asthma and/or asthma medications taken).

In 2019, another secondary analysis of the SNAP trial was designed to investigate whether the absence of infants’ developmental impairments at 2 years was associated with maternal smoking status measured at different point in the trial [26]. After adjusting for some potential confounding factors which are known to be associated with infant development, this study did not report a significant association between measures of maternal tobacco smoke exposure and infant development impairment at age 2. The analyses were adjusted for maternal socioeconomic status, maternal education, low birth weight, maternal exposure to passive smoking, and maternal obesity. The authors concluded that there is no evidence to support the hypothesis that better infant development observed among infants born to women randomized to nicotine patch resulted from smoking cessation induced by nicotine patch use.

### 3.3. Attention-Deficit/Hyperactivity Disorder (ADHD)

In 2014, Zhu et al. published a study about 84,803 children who participated in the Danish National Birth Cohort and followed up to 8 to 14 years old [28]. The information on parental smoking and maternal use of NRT was reported from interviews during pregnancy as in Milidou et al. [24]. The women were asked (self-reported questionnaire) around 16 weeks of gestation if they smoke and/or use NRT (chewing gum, patches, or sprays) and if the fathers smoke. The children with ADHD were identified by a combination of ADHD medication and hospital diagnosis after the age of 5 (from the International Classification of Diseases, 10th revision) from the register of medicinal product statistics and the Danish national Patient register and the Danish psychiatric central register, respectively. Furthermore, the parent reported strengths and difficulties questionnaire (SDQ) was filed out when the children were 7 years old. In this cohort, 2009 (2.4%) children had an ADHD diagnosis or ADHD medication. Among the included children, 240 (0.28%) had mothers using NRT and smoker fathers, and 574 (0.67%) had mothers using NRT and nonsmoker fathers. In comparison with children of nonsmoking parents (reference group), children of mothers using NRT and nonsmoker fathers had higher risk of ADHD (Hazard ratio [HR], 2.28; 95% CI, 1.48–3.51). Maternal exposition to NRT during pregnancy was not statistically associated with ADHD in children in cases of smoker fathers (HR, 1.28; 95% CI, 0.67–2.89). These analyses were adjusted for maternal age, parity, alcohol intake during pregnancy, parental socioeconomic status, parental psychopathology, and child’s gender. Regarding the results about hyperactivity/inattention scores as the outcome, the use NRT during pregnancy was statistically associated with higher hyperactivity/inattention score in 7 years children in comparison with nonsmoker parents (adjusted regression coefficient ß, 0.39; 95% CI, 0.06–0.73 in cases of smoker fathers, and adjusted regression coefficient ß, 0.45; 95% CI, 0.20–0.70 in cases of nonsmoker fathers). The association was stronger for maternal smoking than for NRT using and/or paternal smoking. These results were also adjusted for maternal age, parity, alcohol intake during pregnancy, parental socioeconomic status, parental psychopathology, and child’s gender.

### 3.4. Major Congenital Anomalies

In 2015, Dhalwani et al. reported on a pregnancy cohort from the Health Improvement Network (THIN), a database of prospectively collected data from 570 general practices across the United Kingdom [27]. The pregnancy cohort included all women in THIN aged 15 to 49 years. They included live births between January 2001 and December 2012 and aimed to assess the relationship between early pregnancy exposure to NRT or smoking with major congenital anomalies. Major congenital anomalies were identified in THIN by codes mapped to the European surveillance of congenital anomalies classification system according to the international classification of diseases, 10th revision. The average length of registration data in THIN were up to 5 years. Minor congenital anomalies and anomalies specifically attributed to known teratogens were excluded from the study population. To assess the NRT exposure, the prescriptions for NRT were extracted by using codes in the British national formulary. The NRT group was defined for women with NRT prescribed during the first trimester of pregnancy or within 4 weeks before their estimated conception date. The smoking status of women was assessed by smoking status codes from only first trimester. The absolute risks of major congenital anomalies in the NRT group (women prescribed NRT during the first trimester or 1 month before conception) were compared with those of women who smoked during pregnancy and with a control group (women who neither smoked nor were prescribed NRT). Among 192,498 live-born children, the prevalence of major congenital anomalies was 288 per 10,000 live births (5535 children with at least one major congenital anomaly). Compared with the control group, adjusted OR for major congenital anomalies in the NRT and the smokers group were 1.12 (99% CI, 0.84–1.48) and 1.05 (99% CI, 0.89–1.23), respectively. The results were adjusted for maternal age at conception, Towsend deprivation index score, maternal diabetes, asthma, maternal mental illnesses, and multiple births. The OR comparing the NRT exposed children with children of smoking mothers was 1.07 (99% CI, 0.78–1.47). No statistically significant associations between maternal NRT exposure and system-specific anomalies have been found with the exception of respiratory anomalies (OR 4.65 [99% CI, 1.76–12.25]; absolute risk difference: 3 per 1000 births); however this difference was based on only 10 exposed cases.

## 4. Discussion

This review provides an overview of human child health outcomes after NRT during pregnancy. It shows an association between exposition to NRT during pregnancy and infantile colic among the 6 months old in comparison with no exposition (no smoking). An association is also reported between exposition to NRT during pregnancy and attention-deficit/hyperactivity disorder in children after 5 years old in comparison with nonsmoking (in mothers and fathers). NRT exposure does not seem to be associated with infant behavioral impairment (based on responses to selected questionnaires sent to participants, with items including the ASQ-3); in comparison with placebo even a somewhat better outcome has been reported [25,26,30]. According to one report, NRT exposure during pregnancy was not associated with major congenital anomalies in comparison with a control group (no NRT and no smoking during pregnancy) [27].

The results about infantile colic need to be confirmed because of the limitations of the related study [24]. Hence, Milidou et al. have assessed their primary outcome and the exposure from computer-assisted telephone interviews with possible selection and measurement bias. The definition of their primary outcome could also be discussed.

The data about major congenital anomalies seem reassuring about the safety of NRT during pregnancy, with moderate level of evidence but with a large sample size [27]. However, definitive conclusions cannot be drawn due to the potentially harmful effect of NRT on pulmonary development and potential residual confounding.

The association between NRT exposure during pregnancy and ADHD need also to be confirmed because of the limitations of the study of Zhu et al. [28]. As in Milidou et al., the information about parental smoking and maternal NRT using was self-reported and the number of mothers using NRT was small and so the estimates had wide confidence intervals. Furthermore, some confounding factors could not be explored such as genetic and postpartum caring factors.

Sudden infant death syndrome (SIDS) is strongly and exposure dependently associated with maternal tobacco smoking during pregnancy [31,32,33]. This systematic review was not able to identify papers assessing the association of NRT use in pregnancy and SIDS. Future research should evaluate whether there is or there is not an association between prenatal NRT use and SIDS, one of the most severe postnatal infant outcomes.

The SNAP RCT provided the highest level of evidence about infant outcomes at the age of 2 years old [25,26]. The major limitation of this study is the very low compliance rates of both the nicotine (7.2%) and the placebo patches group (2.8%) which makes difficult to attribute any direct association between the study outcomes and exposure.

The most relevant comparison group to study the risk of NRT during pregnancy on child health is questionable. NRT are often prescribed if the pregnant smokers fail to quit without pharmacological help. The main clinical question is whether prenatal exposition to NRT (pharmaceutical nicotine in various galenic forms) is associated with better child health outcomes than prenatal smoking exposure (nicotine plus other toxins of tobacco) and secondarily whether NRT during pregnancy increases the risk of postnatal child outcomes compared to those children whose mother did not smoke during pregnancy.

Nicotine easily crosses the placental barrier, and in humans it can be detected in the fetal circulation at levels exceeding its maternal concentrations by 15%; amniotic fluid concentrations of nicotine are 88% higher than maternal plasma nicotine concentrations [34,35]. This transfer is rapid with peak plasma concentrations in the fetus after 15–30 min. Nicotine’s metabolism is largely reduced in fetal tissues in particular in the liver that may lead to the increased and long-lasting nicotine concentration in both fetal blood and amniotic fluid [36]. Nicotine’s metabolism is accelerated in pregnant women [37], suggesting that pregnant smokers may need higher daily dose of nicotine substitution than non-pregnant women. Some authors speculate that if higher doses of nicotine are needed during pregnancy to successfully quit smoking, this may contribute to the fetal toxicity of NRT described in animal studies, and can even contribute to the increased risk of attention deficit hyperactivity disorder associated with maternal smoking during pregnancy [28,38]. Animal studies have reported association between nicotine administration and abnormal lung development, leading to structural and functional alterations that persist into adulthood [39,40]. However, increased nicotine intake from NRT during pregnancy is an adaptative behavior to compensate for the increased nicotine metabolism leading to increased need for nicotine substitution; this cannot be assimilated to the observed toxicity of nicotine administered in laboratory conditions.

In 2013, the protocol of an Australian population-based cohort study using linked administrative data has been reported, the Maternal Use of Medications and Safety (MUMS) Smoking Study [7]. This study will explore the utilization, effectiveness and safety of pharmacotherapies for smoking cessation during pregnancy and postnatal outcomes, and the corresponding results would probably be published soon.

It is necessary for the future studies to also include a control group of women with no exposure to nicotine. This would make it possible to assess the potential role of pure nicotine on the health of the offspring. These future studies should be cohorts with precise control groups, with repeated assessment of child health, and with adjustment of confounding factors such as smoking, use of NRT in postnatal period, breastfeeding, second-hand smoke exposure, toxic substances consumption, or maternal psychiatric disorders. Furthermore, all future therapeutic trial dealing with pharmacotherapy smoking cessation should include children long-term follow-up, and use smoking biological markers exposure to be more objective. The combination of NRT, which could be associated with higher odds of smoking cessation [41] should also be studied. Data are also needed about NRT during breastfeeding and child outcomes.

The emergence of new nicotine containing products such as electronic cigarettes and other nicotine delivery systems represent new challenges. These products are often marketed as a safer option than smoking. Data about their efficacy and harm in pregnancy are lacking as do data about health outcome in children whose mothers used them during their pregnancy [17,18].

### Strengths and Limitations

To the best of our knowledge, this is a first systematic review about fetal NRT exposure during pregnancy and postnatal health outcomes. The main limitation of this review is inherent to the paucity of existing data and the very few numbers of studies. The level of evidence of half of the studies is, at best moderate.

Prenatal exposure to NRT does not seem to be associated with infant neurodevelopmental impairment but the only available evaluation with ASQ was done at one point, at two years of age. Repeated, prospective ASQ assessments are needed to identify the neurodevelopmental pattern. Moreover, the parents involved in the RCT SNAP trial may have under- or overestimated the capabilities of their child. It seems that potential social and cultural differences between parents are not factors deemed to alter the validity of their neurologic assessment of their child [42]. However, prospective trials should control for a large number of known potential confounders and apply data analysis models that may take into account the role of unknown confounders.

Conducting studies on the postnatal effect of prenatal NRT exposure remains methodically difficult because of the various galenic forms of NRT and the low rate of smoking cessation. One also may question the relevance of an intention-to-treat analysis in studies with low rates of adherence to NRT.

## 5. Conclusions

This review provides an overview of child health outcomes after exposition to NRT during pregnancy. Low to moderate evidence suggests that NRT use may be associated with higher risk of infantile colic at 6 months similar to that associated with prenatal maternal smoking during pregnancy. NRT during pregnancy may be associated with risk of attention-deficit/hyperactivity disorder as maternal smoking. No association between NRT during pregnancy and infant neurodevelopmental impairment or major congenital anomalies has been shown. The small number of included studies and the large number of cofounding factors do not allow to bring any final conclusions. Well-designed controlled clinical trials are needed to provide more information on the postnatal outcomes regarding the use of NRT and other pharmacotherapies for smoking cessation during pregnancy.

## Figures and Tables

**Figure 1 ijerph-18-04004-f001:**
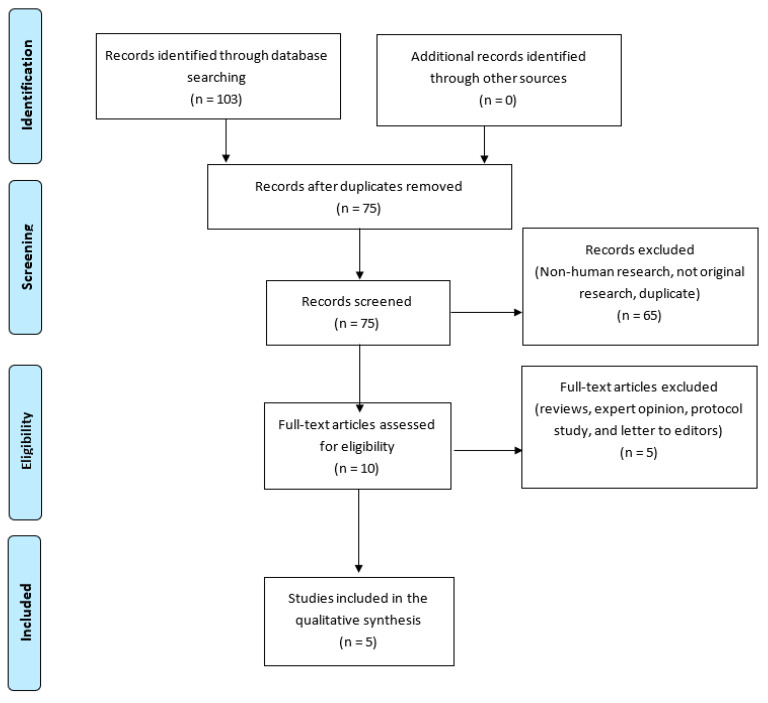
The preferred reporting items for systematic review and meta-analysis (PRISMA) flowchart of the selected studies.

**Table 1 ijerph-18-04004-t001:** Summary of the studies included in the review (superscripts in the table refer to details about outcomes and type of study).

Authors	Year of Publication	Country	Type	Sample Size	Period of Inclusion	Main Outcomes	Secondary Outcomes
Milidou et al. [24]	2012	Denmark	Population-based cohort study	63,128 mother-infant dyads	1996–2002	Infantile colic ^1^	-
Cooper et al. [25]	2014	The Midlands and North-West England	Multicenter RCT “SNAP Trial”	888 infants	1 May 2007 to 1 February 2010	Survival without developmental impairment ^2^	Infant respirator symptoms, smoking outcomes
Zhu et al. [28]	2014	Denmark	Population-based cohort study linked to national registers	84,803 children	1996–2002 (pregnancy) and 7-year follow-up	Attention-deficit/Hyperactivity disorder ^3^	-
Dhalwani et al. [27]	2015	United Kingdom	Retrospective cohort from prospectively collected data (The Health Improvement Network) with linked mother-child primary care records	192,498 live-born children and 5535 children with at least 1 Major congenital anomalies	January 2001 to December 2012	Major congenital anomalies ^4^	-
Iyen et al. [26]	2019	The Midlands and North-West England	Secondary analysis of a Multicenter RCT “SNAP Trial” ^5^	884 infants	1 May 2007 to 1 February 2010	Infant development	-

SNAP, The Smoking, Nicotine and Pregnancy trial. ^1^ Based on the modified Wessel’s criteria: crying or fussing for more than 3 h a day for more than 3 days a week. ^2^ Evaluated by the 24-month participant questionnaire (PQ2) included five domains of the Ages and Stages Questionnaire 3rd edition (ASQ-3) (communication, gross motor, fine motor, problem solving, and personal-social development) and the Health Professional questionnaire (HPQ) designed for completion with medical or health visitor and aiming to measure children’s disability and health. ^3^ Combination of Attention deficit/Hyperactivity (ADHD) disorder medication and hospital diagnosis to identify children with ADHD. ^4^ Major Congenital Anomalies extracted by using Read Codes mapped to the European Surveillance of Congenital Anomalies (EUROCAT) classification system. ^5^ With objective to investigate associations between participants’ smoking measures and infant development (assessed as in Cooper et al.).

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
