# Peer review of "Nicotine Replacement Therapy during Pregnancy and Child Health Outcomes: A Systematic Review"

_ijerph, 2021, doi:10.3390/ijerph18084004_

Round 1
Reviewer 1 Report
Thank you for sending your paper entitled “Nicotine replacement therapy during pregnancy and child 2 health outcomes: a systematic review” to Internacional Journal Environmental Research and Public Health. After carefully review this interesting paper, the following comments are listed for your reference:
- Abstract: To increase potential citations, authors should check keywords against those recommended in the MeSH Browser of Medical Subject Headings https://meshb.nlm.nih.gov/search. For example: “Nicotine replacement therapy”; “congenital anomalies” and “attention-deficit/hyperactivity disorder” are not MeSH. I recommend that you change these keywords.
- Introduction (lines 42-43): the bibliographic citation of this data is missing " Prevalence of maternal smoking ranges between 10.9% and 38.4% in Europe…”
- Methods (line 81): on what date did the search in electronic databases begin?. This information should be added.
- Result: review table 1, I will cut some words.
- Discussion (lines 349-350): as far as I know there is a systematic review registered in "PROSPERO". Have you registered your systematic review?
I attach information:
Ravinder Claire, Sue Cooper, Jo Leonardi-Bee, Tim Coleman, Mary-Ann Davey, Catherine Chamberlain, Ivan Berlin. Pharmacological interventions for promoting smoking cessation during pregnancy. PROSPERO 2019 CRD42019140195. Available from: https://www.crd.york.ac.uk/prospero/display_record.php?ID=CRD42019140195
6. References: check bibliographic citation 21, it seems incomplete.
Author Response
Reviewer 1
Thank you for sending your paper entitled “Nicotine replacement therapy during pregnancy and child 2 health outcomes: a systematic review” to Internacional Journal Environmental Research and Public Health. After carefully review this interesting paper, the following comments are listed for your reference:
- Abstract: To increase potential citations, authors should check keywords against those recommended in the MeSH Browser of Medical Subject Headings https://meshb.nlm.nih.gov/search. For example: “Nicotine replacement therapy”; “congenital anomalies” and “attention-deficit/hyperactivity disorder” are not MeSH. I recommend that you change these keywords.
Response: We would like to thank the reviewer for this suggestion. We’ve changed the keywords with: “Smoking cessation agents”, “tobacco use cessation devices”, “congenital abnormalities”, “Attention deficit disorder with hyperactivity”.
- Introduction (lines 42-43): the bibliographic citation of this data is missing " Prevalence of maternal smoking ranges between 10.9% and 38.4% in Europe…”
Response: We are sorry for this oversight and add the corresponding reference.
- Methods (line 81): on what date did the search in electronic databases begin?. This information should be added.
Response: We have specified line 86: “from the inception of each database”.
- Result: review table 1, I will cut some words.
Response: We are sorry, after new reading of Instructions for authors, we do not understand what is the reviewer’s expectation as to Table 1? We now are using smaller fonts.
- Discussion (lines 349-350): as far as I know there is a systematic review registered in "PROSPERO". Have you registered your systematic review?
I attach information:
Ravinder Claire, Sue Cooper, Jo Leonardi-Bee, Tim Coleman, Mary-Ann Davey, Catherine Chamberlain, Ivan Berlin. Pharmacological interventions for promoting smoking cessation during pregnancy. PROSPERO 2019 CRD42019140195. Available from: https://www.crd.york.ac.uk/prospero/display_record.php?ID=CRD42019140195
Response : We thank the reviewer 1 for this comment. We know this registered systematic review and one of our co-author (IB) was involved in it. However, this systematic review has dealt pharmacological intervention during pregnancy and not with their effect on the child after pregnancy. It has also been reference in the original manuscript as reference 10.
We have not registered our systematic review since we did not plan to run a meta-analysis.
- References: check bibliographic citation 21, it seems incomplete
Response: We thank the reviewer 1 for this comment and we have completed the reference 21.
Reviewer 2 Report
This manuscript provides overview of 5 epidemiologic studies on the possible effect of maternal NRT use during pregnancy on child development and congenital anomality. The effects on development included colic, behavior, and ADHD.
Literature selections and extractions seem to be like systematic review.
Majority of the manuscript was devoted to the description of literatures and discussion is superficial. The manuscript was not successful in providing specific criticisms of the literatures that will be meaningful for future studies. It provides only general criticisms/suggestions. Probably this failure was due to the fact that this manuscript focused on exposure, i.e., NRT, but not on specific outcome(s) the NRT is related with.
Thus, this manuscript is more like a simple review article, that provides an overview of the available epidemiologic literatures in some details, than is a systematic review.
I found some utility as a NRT-related epidemiologic literature summary. But I strongly suggest the authors to remove “systematic review” from the title and text.
Author Response
Response: We thank the reviewer 2 for this analysis and her/his comments. Although we did not register the protocol into Prospero, which is recommended but not a condition to running a systematic review, we rigorously followed the methodology of a systematic review (PRISMA guidelines) as described on page 2 of the manuscript. The literature search catching a large period from inception to December 2020 has provided only a surprisingly low number of papers. However, the number of papers we identified is not, by any means, associated with the search methodology used but with the low level of research interest in prenatal NRT exposures’ postnatal effects. We therefore prefer keeping the wording of “systematic review” in the title and all over the manuscript.
Reviewer 3 Report
I cannot recommend acceptance of the paper in the present state.
The number of studies included is too small and number of cofounding factors is too large to bring any relevant conclusions.
Specific comments and suggestions are given below.
Line 77: Please add exclusion criteria in Materials and Methods section.
Line 82: Why did not authors use more keyword, e.g. ‘’nicotine replacement therapy’’, ‘’smoking cessation’’…?
There is no data of breastfeeding that is also very important factor. Also, the important factor is exposure of the child to tobacco smoke in early childhood: one cannot be sure that impairments are caused exclusively by NRT used in pregnancy. Also, pregnant women very often underreport active smoking. Therefore, there is a need for using biological markers of exposure to be more objective.
Author Response
Comment 1: Line 77: Please add exclusion criteria in Materials and Methods section.
Response: We agree with the reviewer that the number of studies we could include is surprisingly small and they address various outcomes. These findings did not allow drawing firm conclusions about NRT’s potential effects on postnatal health outcomes in the child. We think the issue is important and we hope that this manuscript if published, will contribute to draw the attention of researchers of the field to address this research question.
We thank the reviewer 3 for this comment. We added: “The exclusion criteria were: non-human research, review/meta-analysis/summary, opinion/letter/response, case reports, recommendations, articles with no data on the child health, and articles reporting only data on birth and neonates (before 28 days).” (now lines 93-96)
Comment 2: Line 82: Why did not authors use more keyword, e.g. ‘’nicotine replacement therapy’’, ‘’smoking cessation’’…?
Response: We have used the MeSH term “Tobacco use cessation devices”, this MeSH heading allowed us to explore the following entry terms “nicotine replacement therapy” and “smoking cessation”.
Comment 3: There is no data of breastfeeding that is also very important factor. Also, the important factor is exposure of the child to tobacco smoke in early childhood: one cannot be sure that impairments are caused exclusively by NRT used in pregnancy. Also, pregnant women very often underreport active smoking. Therefore, there is a need for using biological markers of exposure to be more objective.
Response: We thank the reviewer 3 for this interesting comment. We chose to investigate the use of nicotine replacement therapy during pregnancy as the main exposure factor. The included studies do not allow to provide data about all these confounding factors. However, we add in our discussion line 346 the breastfeeding to the suggested confounding factors to use in adjustment in future studies. We also add the suggestion to use smoking biological markers in future studies lines 346-347.
Round 2
Reviewer 1 Report
Accept in present form
Author Response
We thank the reviewer 1 for her/his conclusion.
Reviewer 3 Report
The number of studies included is too small and number of cofounding factors is too large to bring any relevant conclusions.
Author Response
We agree with the reviewer that the number of studies we could include is surprisingly small and they address various outcomes. These findings did not allow drawing firm conclusions about NRT’s potential effects on postnatal health outcomes in the child. We think the issue is important and we hope that this manuscript if published, will contribute to draw the attention of researchers of the field to address this research question. We add in the conclusion lines 430-431: "The small number of included studies and the large number of cofounding factors do not allow to bring any final conclusions."
This manuscript is a resubmission of an earlier submission. The following is a list of the peer review reports and author responses from that submission.